# An Open-Source, Durable, and Low-Cost Alternative to Commercially Available Soil Temperature Data Loggers

**DOI:** 10.3390/s22010148

**Published:** 2021-12-27

**Authors:** Salvatore R. Curasi, Ian Klupar, Michael M. Loranty, Adrian V. Rocha

**Affiliations:** 1Department of Biological Sciences and the Environmental Change Initiative, University of Notre Dame, Notre Dame, IN 46656, USA; ianklupar@gmail.com (I.K.); arocha1@nd.edu (A.V.R.); 2Department of Geography and Environmental Studies, Carleton University, Ottawa, ON K1S 5B6, Canada; 3Climate Research Division, Environment, and Climate Change Canada, Victoria, BC V8W 2Y2, Canada; 4Department of Geography, Colgate University, Hamilton, NY 13346, USA; mloranty@colgate.edu

**Keywords:** Arduino, data logger, DIY, low-cost, microclimate, open-source, soil temperature

## Abstract

Soil temperatures play an important role in determining the distribution and function of organisms. However, soil temperature is decoupled from air temperature and varies widely in space. Characterizing and predicting soil temperature requires large and expensive networks of data loggers. We developed an open-source soil temperature data logger and created online resources to ensure our design was accessible. We tested data loggers constructed by students, with little prior electronics experience, in the lab, and in the field in Alaska. The do-it-yourself (DIY) data logger was comparably accurate to a commercial system with a mean absolute error of 2% from −20–0 °C and 1% from 0–20 °C. They captured accurate soil temperature data and performed reliably in the field with less than 10% failing in the first year of deployment. The DIY loggers were ~1.7–7 times less expensive than commercial systems. This work has the potential to increase the spatial resolution of soil temperature monitoring and serve as a powerful educational tool. The DIY soil temperature data logger will reduce data collection costs and improve our understanding of species distributions and ecological processes. It also provides an educational resource to enhance STEM, accessibility, inclusivity, and engagement.

## 1. Introduction

Temperature plays an important role in the physiology, activity, and distribution of organisms across the globe [1,2,3,4,5,6,7,8]. Temperatures vary across time and space and can substantially differ between the air, water, and soils. Although air temperatures have been extensively measured using global networks or satellites, there has been less attention towards soil and water temperatures [3,6,9,10]. This is unfortunate because many biological and biogeochemical processes rely on proximate temperatures in soil and water that can differ widely from those in the air [10,11,12,13,14,15]. In soils, temperatures can be difficult to predict from those in the air because of their dependence on radiation load, surface energy budget partitioning, soil depth, and soil thermal properties that vary with soil texture and moisture content [12,13,14,16,17,18]. Hence, adequately characterizing the three-dimensional variability in soil temperatures across a site often requires a large number of sensors that can be costly to purchase and deploy.

Do it yourself (DIY) soil temperature data logger systems are a potential solution to the high cost of deploying a large number of sensors. DIY approaches can reduce overhead, labor, consumable, and production costs; resulting in a more cost-effective data logger system that can be scaled towards greater measurement capability (i.e., an economy of scale) [19,20,21,22,23]. The downsides of DIY approaches are that they often require proper training and educational background that may limit their applicability in the context of ecological research. However, current technological advances and online communities have reduced these limitations by increasing the accessibility of information through online lectures or step-by-step video tutorials [22]. Online resources have the potential to close the “skills gap” that prevents the implementation of DIY data loggers while also allowing them to serve as powerful educational tools.

Here we develop an open-source soil temperature data logger based on the Arduino microcontroller platform. We also develop online resources and a tutorial video series to allow people with little electronics experience to implement this DIY soil temperature data logger system. Finally, we test DIY soil temperature data loggers constructed by students in the field in Alaska and in the lab against a commercial system. Our DIY soil temperature data logger can serve as both a powerful scientific and educational tool. It can provide data necessary for increasing our understanding of the spatial and temporal variability in soil temperature while also helping to train the next generation of scientists and engineers.

## 2. Materials and Methods

We developed several criteria to construct an accessible DIY soil temperature data logger system. The DIY system needed to be inexpensive, capable of long deployments without battery changes or data retrieval, easy to construct, and rugged enough to be deployed in extreme environments. With these criteria in mind, we designed a soil temperature data logger system based on the Arduino platform which utilizes digital (DS18B20, Maxim Integrated, San Jose, CA, USA) temperature sensors. Arduino is an open-source hardware and software company, project, and user community (https://www.arduino.cc/, Last accessed: 1 December 2021). It provides hardware kits, software (i.e., an API and libraries), and educational resources (i.e., tutorials and Q&A forums). We chose to use a printed circuit board design rather than off-the-shelf modules or solderless breadboards to reduce cost and ensure the final device was robust. Then we created a suite of online resources intended to ensure that this design is broadly accessible even to those with little prior electronics knowledge. These resources included a detailed series of instructional videos (https://www.youtube.com/channel/UC5EXX-9zh4DVggihpJ8SJuQ, Last accessed: 1 December 2021), along with written instructions, a parts list, and software meant to simplify device setup and testing (https://github.com/RochaLabND/SoilTemperatureLogger, Last accessed: 1 December 2021). All of these resources were also archived on Zenodo (https://doi.org/10.5281/zenodo.5781439, Last accessed: 1 December 2021) to ensure they remain accessible. These materials were used to train 15 undergraduate and four graduate students to build and test the DIY data logger systems in the field.

A detailed description of the materials used to build the DIY soil temperature data loggers can be found in the youtube video series and on GitHub. The DIY soil temperature data logger consisted of a custom printed circuit board that was controlled by an embedded microcontroller (ATMEGA328P, Microchip Technology, Chandler, AZ, USA) running open-source Arduino software (Figure 1A). The onboard peripherals included a battery-backed real-time clock (DS1307N+, Maxim Integrated, San Jose, CA, USA), and onboard logic level shifting circuitry that allowed for the implementation of a removable Secure Digital (SD) card for data storage. The board included a high-efficiency switched-mode power supply (U1V11F5, Pololu Corporation, Las Vegas, NV, USA) and circuitry with an additional microcontroller (PIC12f683, Microchip Technology, Chandler, AZ, USA), to place the data logger into an ultra-low power state between measurements. The board included holders for two lithium AA batteries as well as a single coin cell battery for the real-time clock. The board included a total of 11 ports for internal (x1) and external (x10) digital temperature sensors (DS18B20, Maxim Integrated, San Jose, CA, USA). The digital temperature sensors do not require additional hardware (i.e., an analog to digital converter, voltage reference, and/or multiplexer) to convert analog measurement signals from the sensors to a digital output.

As described in the online tutorials and videos, the construction of the data loggers occurred in four phases: (1) part procurement, (2) assembly, (3) programming, and (4) testing. The custom printed circuit boards were procured from a low volume PCB manufacturer using CAD files (Seeed Technology Inc. Shenzhen, China). The electronic components were procured from online distributors (Mouser Electronics Mansfield TX and Digi-Key Electronics Thief River Falls, MN, USA). Data loggers were assembled by undergraduate and graduate students at the University of Notre Dame with little electronics experience utilizing commonly available tools (pliers, side cutters, and soldering irons) and our online instructional materials. Data loggers were programmed utilizing our code provided on GitHub, and the applicable USB interfaces (FTDI Friend, Adafruit Manhattan, NY; PICKIT 3, Microchip Technology, Chandler, AZ, USA). Interactive software routines were used to register the external temperature sensors with the data logger’s software each on an independent channel. Finally, an interactive self-testing routine was utilized to ensure the proper functioning of the data logger. Following testing, the data loggers were waterproofed by placing them into inexpensive enclosures consisting of an approximately one-foot length of PVC pipe capped at both ends (Figure 1B,C). Cable glands were installed at the end of the pipe to allow the external temperature sensors to pass through the enclosure.

The durability and performance of the DIY data logger system were evaluated with a three-year field deployment on the North Slope of Alaska and a laboratory inter-data logger system cross-validation with an Onset HOBO Pro v2 external temperature data logger (U23-003). We chose the Onset HOBO Pro v2 mainly due to the similarity in design and deployment ability between these systems. For the field test, the DIY data logger system was deployed in the arctic where low temperatures, remoteness, and high animal activity make field measurements challenging. DIY data logger systems were installed on the North Slope of Alaska at the Toolik Lake Long Term Ecological Research (LTER) station. A total of 14 temperature data loggers each set up to log data from four external temperature sensors and one internal temperature sensor were deployed. The sites were revisited annually to download the data, but no preventive maintenance or repairs were performed on the data loggers. At Toolik Lake, air temperatures can be less than −45 °C in the winter. The mean annual air temperature is −7 °C, the mean growing season (June-August) temperature is 6 °C, and the mean non-growing season (September-May) temperature of −11 °C. The mean annual precipitation at Toolik Lake is 318 mm with 40% occurring as rain and 60% occurring as snow [24,25]. The inter-data logger system cross-validation was conducted in the lab using a water bath, a hot plate, and a −20 °C freezer. For testing at temperatures greater than 0 °C, both the HOBO and DIY temperature sensors were placed in the same water bath and gradually heated to 70 °C over 6 h using a hot plate. For testing at temperatures less than 0 °C, both the HOBO and DIY temperature sensors were placed in the same water bath in a −20 °C freezer for 12 h. Temperature data were logged every minute and later aggregated to 30-min averages to minimize thermal disequilibrium between the sensor and environment and differences in response time of the two data logger systems.

The performance specifications and cost of the DIY sensors were compared to two commonly used soil temperature systems from Campbell Scientific and Onset Corporation. For each manufacture, we determined the most cost-effective data logger and sensor configuration that maximized measurements per unit cost. For Campbell Scientific, the most cost-effective data logger configuration consisted of three stacked AM16/32 multiplexers connected to a CR800 to make measurements and store data. The Campbell scientific single-ended 108 temperature probe sensed soil temperature, and the data logger and sensor system were powered by a 12 V 12 Ah battery connected to a 10 W solar panel. For Onset Corporation, the most cost-effective configuration was the HOBO Pro v2 external temperature data logger (U23-003) that included two soil temperature sensors powered by an internal 3.6 V battery. For our DIY soil temperature data logger, the most cost-effective configuration was an enclosed data logger with 10 external sensors. Performance specifications for each system were obtained through online manuals and compared among the three data logger systems. In comparing costs, we included labor costs for the DIY data logger systems based on a minimum wage (i.e., 15 USD h^−1^) with labor time determined by the length of the youtube video series (~2 h). This provided a liberal estimate of labor cost since construction time will decrease to 30–45 min once the user becomes familiar with the protocol. One-time costs were also calculated for each system and included manufacturing tools, data transfer cables, and proprietary software. In calculating the one-time costs for the DIY soil temperature data logger we assumed that high-quality tools would be used. Lastly, the cost of supporting 1–200 sensors for the HOBO, Campbell, and DIY sensors was calculated taking into account how costs step as additional peripherals and sensors are added to determine savings attained from the economy of scale.

## 3. Results

Despite having almost no prior background in electronics, graduate and undergraduate students were able to construct functional soil temperature data loggers using our tutorials (Figure 1). The DIY soil temperature data loggers had similar technical specifications as other commonly used data logger systems (Table 1). Data logger and sensor operational ranges, accuracy, resolution, response time, and drift differed by less than 10–30% among the three systems (Table 1). However, the three data logger systems largely differed in their battery life and data storage capacity. The Campbell system had a longer battery life due to its use of a solar panel, while data storage capacity was several orders of magnitude higher for the DIY system due to its use of commercially available SD cards.

The DIY data logger system performed similarly to other systems on the market in our lab and field tests (Figure 2). For example, the lab inter-comparison between the DIY and HOBO data logger systems revealed strong agreement across a wide range of temperatures with an R^2^ of 0.99, a slope of 1.0, and an offset of −0.5 °C (Figure 2A). The percent mean absolute error between the two data logger systems was temperature-dependent and was 2% from −20–0 °C and 1% from 0–20 °C. There was strong agreement in the temperature observations from the DIY data logger systems individual external probes during this test. Across a wide range of temperatures the R^2^ was 1 with a slope of 1, an offset of 0.02 °C, and a residual standard error of 0.18 °C. The percent mean absolute error was 0.6%. Once installed in the field, a majority of the DIY sensors collected uninterrupted soil temperature data across five depths over 3 years (Figure 2B). These DIY sensors produced the expected spatial and temporal patterns with a decrease in the annual average and seasonal cycle of soil temperature with depth. The durability of the DIY sensors also was high with a <10% failure rate in the first year of deployment and a ~50% failure rate by the third year of deployment (Figure 2C); 57% (four units) of the failures were due to water damage, 29% (two units) of the failures were due to battery failure, and 14% (one unit) of the failures were due to animal damage.

The DIY data logger system provided an inexpensive alternative for soil temperature measurement acquisition and storage (Table 2; Figure 3). The three systems varied widely in terms of sensor cost, labor cost, and one-time costs. The number of sensors supported on the Campbell system was an order of magnitude higher than both the DIY and HOBO systems. The labor cost for the DIY sensors was small, while the Campbell and HOBO systems had no labor costs. Because the DIY data logger system requires several basic tools to assemble one-time costs were ten times higher than the Campbell data logger systems but comparable to that for HOBO. The DIY data logger was less expensive than the Campbell and HOBO data loggers when equipped with between one and 200 sensors (Figure 3). Despite having the largest One-time and labor costs among the three systems, the DIY sensors were ~1.7 times less expensive per sensor than the Campbell system and ~7 times less expensive per sensor than the HOBO system in their most cost-effective configurations (Table 2). This low cost per sensor provided large savings for the DIY system when brought to scale; constructing 200 DIY soil temperature sensors saves ~1800 USD over the Campbell system and ~16,000 USD over the HOBO system (Figure 3).

## 4. Discussion

We developed a durable and low-cost DIY data logger system that was comparable in capability and performance to other systems on the market. This DIY data logger system can be easily manufactured by a wide audience using the online tutorial materials available on GitHub and Youtube.com. This low-cost and simple-to construct DIY soil temperature data logger system had several scientific and societal benefits that will improve our ability to characterize soils and enhance STEM education. First, the large cost savings that the DIY soil temperature data logger systems provide at scale increases our ability to characterize the often large vertical and horizontal variations in soil temperature. Characterizing such variation will improve our understanding of the spatial heterogeneity of land-atmosphere fluxes and species distributions and provide better validation of land surface models [3,26,27,28,29,30]. Second, the open-source tutorials, easy to construct design, and link to the robust Arduino community mean the DIY data logger system provides an educational resource to enhance STEM inclusivity and engagement through construction, data acquisition, and confidence-building [31,32,33,34]. Lastly, the low cost of the sensors also increases inclusivity by providing a mechanism for low-income or marginalized populations to participate in environmental monitoring [23,35].

Several tradeoffs existed between our DIY soil temperature data logger and commercial systems from manufacturers like Campbell and Hobo. The potential deployment length of our DIY soil temperature data logger was longer than that of Hobo systems owing to its long battery life and high data storage capacity, but shorter than that of a Campbell system paired with a solar panel. The durability of our DIY soil temperature data logger was comparable to that of Hobo data loggers which can have similar issues with sensors failures due to enclosure leaks and standing water. Campbell systems have more robust hardware and enclosure designs but are less portable due to their weight and bulky peripherals (i.e., solar panels and batteries). Although a Campbell system has the potential to support an extremely larger number of sensors it’s analog design means that wire length limits decrease the spatial extent the sensors can cover. Due to its modular design, the DIY soil temperature logger is particularly well suited to deploying clusters of sensors (i.e., less than 50) at broadly spaced or remote sites. Overall our DIY soil temperature data logger was cost-effective when compared to commercial systems and suitable for developing spatially extensive networks of sensors.

Our field testing has allowed us to identify several areas in which our DIY soil temperature data logger could be improved in future iterations to increase reliability. Commercial data loggers experience failures in the field with reported failure rates of 7–27% per year [20,36,37,38,39]. The majority of our data logger failures were due to leaks. Many of these failures occurred in a relatively short period potentially as a result of a weather event, inundated soil, ponded surface water, or some combination of the three. Decreasing the overall size of our DIY soil temperature data logger would allow it to use an inexpensive and robust off-the-shelf waterproof enclosure helping to address this issue. The sensors probes could be reinforced with high-quality adhesive-lined waterproof heat shrink and the printed circuit boards could be sprayed with conformal coating after assembly to improve their performance in wet and inundated conditions. The data loggers were mounted low to the soil surface so increasing the length of the probes cabling could allow them to be mounted further above the ground to decrease the incidence of animal damage or exposure to ponded water. Finally installing larger batteries, conducting preventative maintenance, and implementing further pre-deployment testing could help address and prevent hardware and battery issues.

Our device is open source meaning that users have access to the underlying design including the code, schematics, and board files. Moreover, the Arduino community provides an array of pre-built hardware and associated software which would be cost-prohibitive to design were it not for this robust community. Users who are familiar with the Arduino platform could interface this hardware with a variety of existing digital sensors. Future iterations could incorporate a high-resolution analog to digital converter and precision voltage reference which would allow the device to collect data from a variety of analog sensors. In the future, the device could also integrate additional connectivity options (i.e., Bluetooth or Wi-Fi connectivity). Ultimately our open-source design provides users with access to the underlying code and hardware ensuring they will be able to address limitations and adapt it to their purposes.

The DIY nature of the device and the need for user assembly allow it to be used as a powerful education tool for students interested in STEM. Its low cost and link to the robust Arduino community mean that it can serve as an entry point into STEM for students from underrepresented groups at a variety of age levels. This work is a step towards increasing access to as well as the spatial resolution of soil temperature monitoring across the globe while simultaneously working to train the next generation of scientists and engineers.

## 5. Conclusions

We developed an inexpensive DIY soil temperature data logger and educational materials to close “skill gaps” that could prevent others from constructing the device. Our results show that students with little electronics experience were able to construct the DIY soil temperature data logger and that the devices were accurate, reliable, and inexpensive when compared to commercial systems. Because the device is open source future development could address limitations identified in our testing and integrate new functionalities into the design. Our DIY soil temperature data logger is well-positioned to satisfy the pressing need to increase the spatial resolution of soil temperature monitoring. It can also simultaneously work as an educational resource to train the next generation of scientists and engineers and enhance STEM, accessibility, inclusivity, and engagement. 

## Figures and Tables

**Figure 1 sensors-22-00148-f001:**
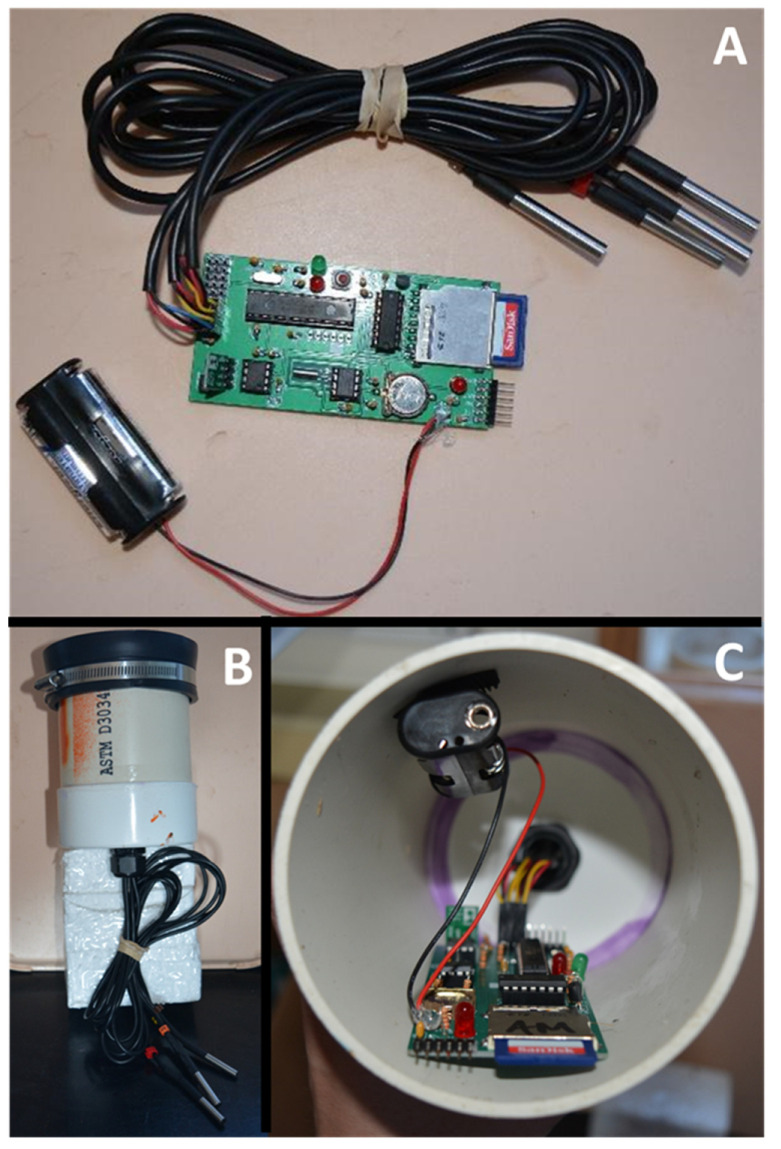
(**A**) A fully assembled do-it-yourself (DIY) soil temperature data logger with four external temperature sensors. (**B**) External view of a DIY soil temperature data logger in an enclosure made from PVC pipe. (**C**) Internal view of a DIY soil temperature data logger mounted in an enclosure made from PVC pipe.

**Figure 2 sensors-22-00148-f002:**
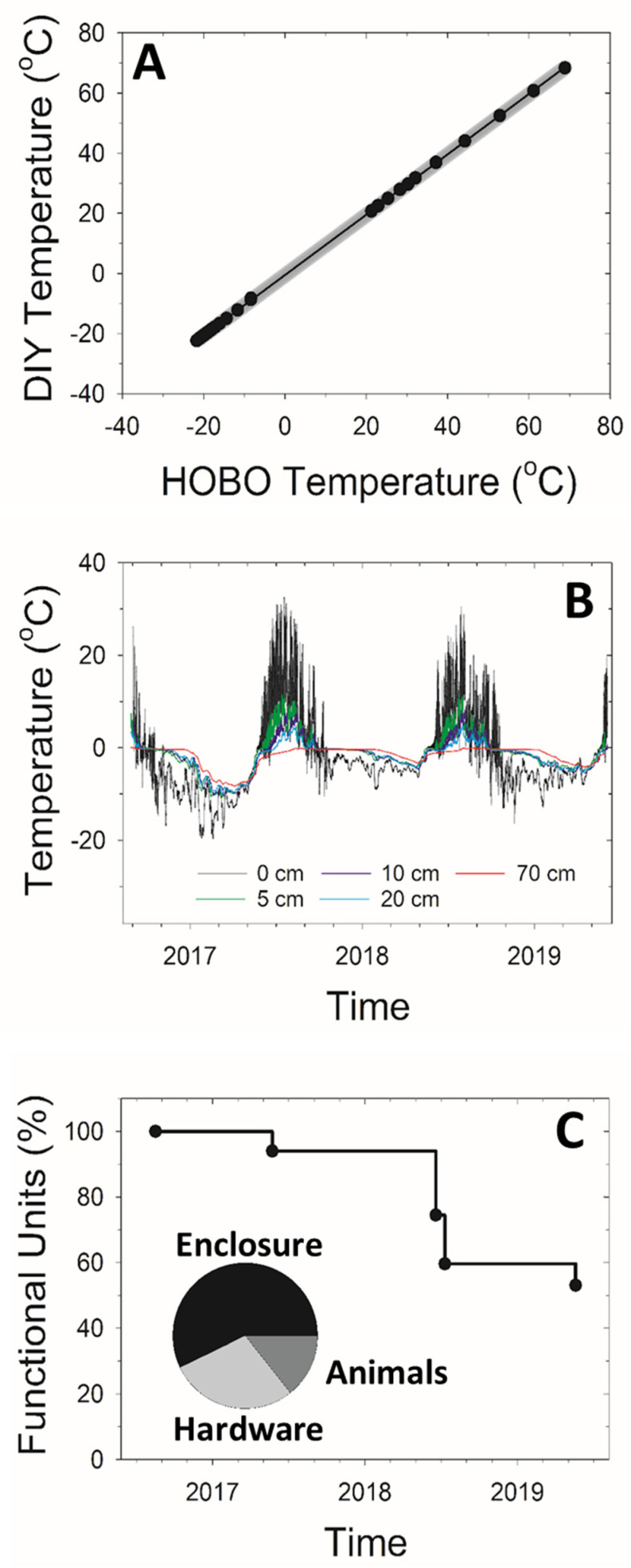
(**A**) Laboratory cross-validation between an Onset HOBO Pro v2 external temperature data logger (U23-003) and the DIY soil temperature data logger. Half hourly averages are shown to minimize thermal disequilibrium and differences in sensor response time. (**B**) Plot of average site temperature at five depths from our field test on the North Slope of Alaska. (**C**) Plot of the percentage of DIY soil temperature data loggers still functional based upon the data recorded versus time from our field test on the North Slope of Alaska. The inset pie chart depicts the causes of data logger failure.

**Figure 3 sensors-22-00148-f003:**
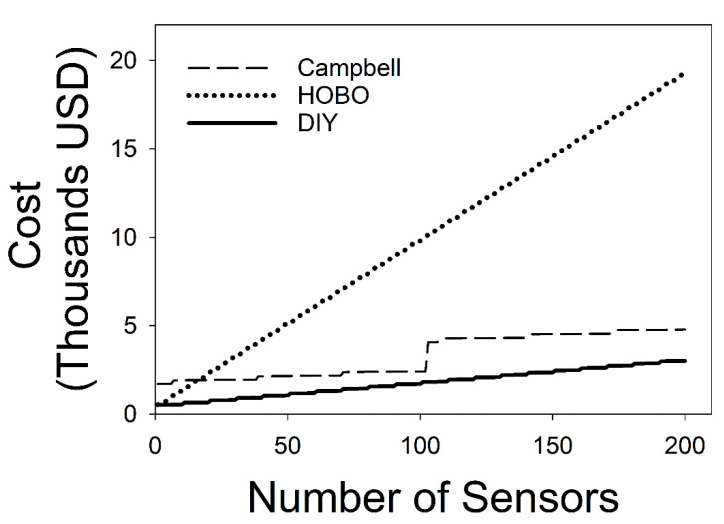
A plot of the cost of supporting 1–200 sensors for HOBO, Campbell, and the DIY soil temperature data logger.

**Table 1 sensors-22-00148-t001:** Technical specification of three soil temperature data logger systems including our DIY soil temperature data logger based upon the manufacturer’s datasheets for the equipment or components.

Specification	Campbell	HOBO	DIY
Sensor Range (°C)	−50–70	−40–70	−55–125
Accuracy (°C from −42–32 °C)	±0.43	±0.30	±0.50
Resolution (°C @ 25 °C)	<0.03	~0.02	0.0625
Data storage (MB)	4	0.064	3000+
Response time in water (s)	<30	30	30
Drift (°C yr^−1^)	<0.1	<0.1	<0.1
Battery life time (yrs)	>4–10	3	6

**Table 2 sensors-22-00148-t002:** Cost of three soil temperature data logger systems including our DIY soil temperature data logger setup in their most efficient configurations. One-time costs refer to reusable manufacturing tools, data transfer cables, and proprietary software.

	Campbell	HOBO	DIY
Sensors supported (#)	102	2	10
Labor (USD sensor^−1^)	0	0	3
One-time cost (USD)	42	340	400
Sensor + data logger (USD sensor^−1^)	23.21	94.5	13

## Data Availability

The materials that support this study are currently available on GitHub at https://github.com/RochaLabND/SoilTemperatureLogger (Last accessed: 1 December 2021) and archived on Zenodo at https://doi.org/10.5281/zenodo.5781439 (Last accessed: 1 December 2021).

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
