# Peer review of "An Open-Source, Durable, and Low-Cost Alternative to Commercially Available Soil Temperature Data Loggers"

_sensors, 2021, doi:10.3390/s22010148_

Round 1

Reviewer 1 Report

Your article has high practical impact, although nothing new was invented as stated by authors. But well tested equipment, great documentation on creating this DIY device is very useful. Making project open source with freely available schematics for every one is great idea, which support overall research merit.

High practical importance of your results is confirmed by long term monitoring with calculation characteristics and financial comparison.

Results can be used in different fields, not only where the authors suggest to use their results. So they will be interesting for wide audience.

Author Response

Thank you for taking the time to review this work and for your insightful comments. The pdf provided here contains our detailed responses.

Reviewer 2 Report

The results look encouraging and motivating. But there are still some contents, which need be revised in order to meet the requirements of publish. A number of concerns listed as follows:

(1)The abstract should be improved. Your point is your own work that should be further highlighted.

(2) In the introduction, the authors should clearly indicate the contributions and innovations of this paper. The main contributions and innovations of this paper can be located at the end of the Section of Introduction.  

(3) The values of parameters could be a complicated problem itself, how the authors give the values of parameters. For example, “USD 15 hr-1”、“air temperatures can reach <-45 °C in the winter with a mean annual 115 air temperature of -7 °C; a mean growing season (June-August) temperature of 6 °C; and 116 a mean non-growing season temperature of -11 °C.”…

(4) In order to highlight the introduction, some latest references should be added to the paper for improving the reviews part and the connection with the literature. For example, https://doi.org/10.1007/s12559-021-09871-4; 10.3390/app11125385; https://ieeexplore.ieee.org/document/9525411 and so on

(4) There are some grammatical mistakes and typo errors. please proof read from native speaker. For example, “soil temperature characterization and prediction requires large and often expensive networks of data loggers.”à“soil temperature characterization and prediction require large and often expensive networks of data loggers”,….

(5) The theoretical background of the proposed method is adequately detailed in the paper.

(6) The conclusion and motivation of the work should be added in a more clear way

(7) More statistical methods are recommended to analyze the experimental results, such as comparative analysis, correlation analysis, variance analysis, ROC analysis and so on.

Author Response

Thank you for reviewing this work and for your insightful comments. The pdf provided here contains our detailed responses to your comments.

Reviewer 3 Report

This article, An open-source, durable, and low-cost alternative to commercially available soil temperature data loggers, describes the design, testing and field deployment of a Do It Yourself (DIY) data logger based on the Arduino platform for logging spatially dispersed soil temperatures.  Extensive documentation and code have been prepared and shared on third party platforms: YouTube, GitHub and Zenodo.  It is refreshing to see equipment design driven by scientific need and in a way that makes it accessible and educational to everyone. 

I liked the potential and was excited to review this article.  I have been involved in field testing of both commercial and DIY instruments, including on the North Slope of Alaska, and can appreciated the effort that went into this project and the ability to record new, unique data sets.  I think with a little more effort this article could be much improved.  I very much like the emphasis on lack of spatially disperse data and the potential to involve students in a STEM activity learning real practical skills.  I was disappointed to see the narrow focus on a specific temperature sensor model and not a more general logger usable with different sensors.  The discussion does touch on future improvements but stating up front that only the one model of sensor is being used would be helpful.  I’d also like to see more data from the testing and inter-comparison of the DIY systems.  It would be nice to know how much variation there is between DIY systems.  My guess is that it is minimal since the sensors’ output is digital and that the overall differences will be similar to the differences between sensors.  I find the comparison of costs far from straight forward.  The cost of the Campbell system might be low if widely dispersed measurements are needed.  Maybe instead of a cost per sensor/measurement approach, just the overall cost for the Toolik deployment using the three different systems.  This would also necessitate a better description of the measurements made at Toolik.  Maybe include a table that compares the pros and cons of each system.  My last comment concerns relying on YouTube and GitHub for much of the documentation and maybe this is a concern for the publisher as well.  I think YouTube and GitHub are excellent to make your work widely available but might be lacking in permanence.  The Zenodo archive seems like a more permanent option and should be mentioned more prominently at the beginning of the paper.  The publisher may also be able to host some of this as supplemental material. 

74-75 Text calls out “YouTube video series” but link is to GitHub repository.

89-106 Might be nice to see the “four phases” split out as bullet points.

107-108: How many sensors/boards were deployed at Toolik?  Were they spatially dispersed in a way that would be difficult to do with the Campbell system?

Table 1: How did you determine the “operational range” of the DIY system?  Specs on the SD card, lithium batteries and temperature sensor are all have a minimum operational temperature of about -40C.

177-180: I think this failure rate is too high.  I’ve had Campbell soil temperature systems run in the Arctic for years with nearly zero failure of the hardware.  Animal damage is by far the biggest problem.  That said I think the failures due to water damage and batteries might have inexpensive fixes.

Figure 2A: I’d like to see more inter-comparisons.  Does each of the DIY system have a similar relationship to the HOBO system? 

Figure 2C: Odd that you have nearly 50% sensors die in just four episodes?  I’m guessing that this plot shows the number of functional sensors when field visits were made?  Might be better to use the data and plot the number of good sensors based on when they have good data.

Figure 3: I’m not sure I like this figure.  I appreciate that the smooth lines are appealing, but I wonder if some detail is lost.  A CR800 by itself could get you 1-6 temperatures.  Adding one AM16/32 can get you another 32 for the one cost and then on in blocks of 32.  Similarly the DIY system should go in blocks of 10. 

Table 2: I think the caption should clearly state that the one-time costs are tools and interfaces needed for building and/or collecting data from the systems.

232-236:  Yes!  I think modularity of the DIY system needs to be a more prominent point. 

261: Add videos to Zenodo.

Author Response

Thank you for agreeing to review this work and for your detailed comments. The pdf provided here contains our detailed responses to your comments.

Round 2

Reviewer 2 Report

ok

Reviewer 3 Report

I think the extra effort put on the polish the article needed.  I recommend for publication.